# A role for cerebellum in the hereditary dystonia DYT1

**Rachel Fremont[†], Ambika Tewari[†], Chantal Angueyra, Kamran Khodakhah\***

Dominick P. Purpura Department of Neuroscience, Albert Einstein College of Medicine, New York, United States

**Abstract** DYT1 is a debilitating movement disorder caused by loss-of-function mutations in torsinA. How these mutations cause dystonia remains unknown. Mouse models which have embryonically targeted torsinA have failed to recapitulate the dystonia seen in patients, possibly due to differential developmental compensation between rodents and humans. To address this issue, torsinA was acutely knocked down in select brain regions of adult mice using shRNAs. TorsinA knockdown in the cerebellum, but not in the basal ganglia, was sufficient to induce dystonia. In agreement with a potential developmental compensation for loss of torsinA in rodents, torsinA knockdown in the immature cerebellum failed to produce dystonia. Abnormal motor symptoms in knockdown animals were associated with irregular cerebellar output caused by changes in the intrinsic activity of both Purkinje cells and neurons of the deep cerebellar nuclei. These data identify the cerebellum as the main site of dysfunction in DYT1, and offer new therapeutic targets.

**\*For correspondence:**
k.khodakhah@einstein.yu.edu

[†]These authors contributed equally to this work

**Competing interests:** The authors declare that no competing interests exist.

## Introduction

Dystonia is a common, debilitating movement disorder caused by co-contraction of agonist-antagonist muscle pairs (*Frucht, 2013*). The underlying neurological causes of dystonia are not fully understood and there are few effective therapeutic interventions. The most common inherited dystonia is early onset generalized torsion dystonia, referred to as DYT1. The onset of symptoms in DYT1 patients occurs before the age of 26 with a focal dystonia that often generalizes, resulting in severe disability (*Bressman et al., 2000*). The majority of DYT1 patients share a single amino acid deletion in torsinA (*Ozelius et al., 1997*; *Risch et al., 1995*; *Warner and Jarman, 1998*), resulting in loss of function of the protein (*Goodchild et al., 2005*). Studies in DYT1 patients have implicated several brain areas in this disorder, including the cerebellum (*Eidelberg et al., 1998*) and basal ganglia (*Panov et al., 2013*). How torsinA disruption causes dystonia and which brain areas play a key role remain outstanding questions.

Recent studies in an animal model of Rapid-onset Dystonia Parkinsonism (RDP) have suggested that in this rare genetic dystonia, abnormal cerebellar output alters basal ganglia activity and likely contributes to dystonic postures (*Calderon et al., 2011*; *Chen et al., 2014*; *Fremont et al., 2014*). Examining the role of these brain regions in a symptomatic model of DYT1 may provide insight into its pathophysiology. Unfortunately, mice with complete knockout of torsinA exhibit early lethality and cannot be studied (*Goodchild et al., 2005*). Further, no viable rodent models of DYT1 show overt dystonia (*Dang et al., 2005*; *Goodchild et al., 2005*; *Grundmann et al., 2012*, *2007*; *Page et al., 2010*; *Sharma et al., 2005*; *Shashidharan et al., 2005*). The same is true of brain region-specific genetic models of torsinA dysfunction (*Liang et al., 2014*; *Pappas et al., 2015*; *Weisheit and Dauer, 2015*; *Yokoi et al., 2011*, *2008*; *Zhang et al., 2011*). While two recently generated mouse models of DYT1 exhibited some twisting movements (*Liang et al., 2014*), these mice do not exhibit the overt sustained postures common in dystonic patients.

**eLife digest** Dystonia is the third most common type of movement disorder after Parkinson's disease and tremor. Patients with dystonia experience prolonged involuntary contractions of their muscles, often causing uncontrollable postures or repetitive movements. Almost thirty years ago, genetic studies revealed that a mutation in the gene that encodes a protein called torsinA causes the most common type of dystonia, called DYT1.

Exactly how mutations that affect the torsinA protein give rise to DYT1 remains unclear, and there are still no effective treatments for the disorder. Part of the problem is that we do not fully understand how torsinA works, or which of its many proposed functions is relevant to dystonia. Moreover, attempts to study DYT1 using genetically modified mice have proved largely unsuccessful. This is because mice that simply express the same genetic mutations that cause dystonia in humans do not show the overt symptoms of dystonia.

Fremont, Tewari et al. have now generated a mouse 'model' that does show symptoms of dystonia, and used these model mice to investigate the role of torsinA in the disorder. Acutely reducing the amount of torsinA protein in a region of the brain called the cerebellum induced the symptoms of dystonia in the mice. Conversely, reducing the amount of torsinA in a different brain area known as the basal ganglia had no such effect, even though both the cerebellum and the basal ganglia contribute to movement. Furthermore, neither manipulation had any effect in juvenile mice, which suggests that, in contrast to humans, young mice can compensate for the loss of torsinA.

Fremont, Tewari et al. also found that the loss of torsinA causes the cerebellum to generate incorrect output signals, which in turn trigger the abnormal movements seen in dystonia. In the future, further studies of the model mice could identify the exact changes that occur in neurons following the loss of torsinA from the cerebellum. Understanding these changes could potentially pave the way for developing effective treatments for DYT1 and other dystonias.

Recapitulating the major symptoms and disease progression seen in DYT1 patients has been challenging in mouse models and therefore it has been difficult to address how loss of torsinA causes dystonia. Genetic models of DYT1 to date involve manipulation of torsinA during embryogenesis. But there is evidence that while there is high embryonic expression of the protein in rodents (*Vasudevan et al., 2006*; *Xiao et al., 2004*), it is only expressed postnatally in humans (*Siegert et al., 2005*). This suggests that compensation of torsinA is highly likely in rodents which could explain discrepancies between current DYT1 rodent models and patients with DYT1.

We hypothesized that regional, acute knockdown of torsinA in the adult mouse, when expression is similar in mice and humans might better replicate DYT1 symptoms. In fact, we previously showed in a mouse model of RDP that acute knockdown of the $\alpha 3$ isoform of the $Na^+/K^+$ - ATPase pump in the cerebellum of adult rodents was sufficient to induce dystonia, whereas genetic mouse models targeting the gene early in development did not produce dystonia. This provides further support for compensatory mechanisms in the rodent following the loss of certain genes early in development. We found that in adult mice, knockdown of torsinA in the cerebellum but not the basal ganglia resulted in severe and persistent dystonia. Symptoms were associated with abnormal erratic cerebellar output caused by dysfunctional intrinsic pacemaking activity of both Purkinje cells and neurons of the deep cerebellar nuclei. In contrast, mice with cerebellar knockdown of torsinA during the early post-natal period had mild symptoms with no dystonia. This observation suggests that indeed compensation of torsinA in early rodent brain development can explain the major differences in phenotype between adults and juveniles after torsinA knockdown.

It is tempting based on these and previous findings (*Fremont et al., 2015*) to hypothesize that abnormal cerebellar activity may be a common mechanism in some hereditary dystonias. In fact, imaging studies have implicated the cerebellum in another inherited dystonia (*Carbon et al., 2010a*, *2013*). Overall, by utilizing acute knockdown in adult rodents, we identified a mechanism whereby disruption of torsinA produces aberrant firing in the cerebellum and results in dystonic symptoms and provide evidence that the cerebellum may initiate symptoms in the most common inherited dystonia, DYT1.

# Results

## Knockdown of torsinA in the basal ganglia does not result in dystonia

Since dystonia is often associated with changes in the basal ganglia, we first explored whether torsinA KD in this brain region results in symptoms. AAVs expressing a shRNA against torsinA with a green fluorescent protein marker (AAV-TorsinAshRNA-GFP, *Figure 1A*) were stereotaxically injected into the basal ganglia (striatum and globus pallidus) of adult mice. Significant knockdown of torsinA (average torsinA levels after KD: 0.187 ± 0.06) was achieved and post-mortem histology demonstrated robust expression throughout the injected areas (*Figure 1B,C*). The behavior of these mice in the open field was assessed by observers blinded to the animals' condition on a previously published dystonia scale (*Calderon et al., 2011*). Since shRNAs take approximately two weeks to be expressed in vivo (*Fremont et al., 2015*), the behavior of the animal can be compared before and after expression of the shRNA. Animals injected with a shRNA against torsinA in the basal ganglia did not develop dystonia, as assessed by the dystonia scale. Moreover, the behavior of the mice was similar to controls injected with a non-targeted shRNA (AAV-NTshRNA-GFP) (*Figure 1D*, *Video 1*). Multiple parameters of open field behavior quantified in animals with knockdown of torsinA in the basal ganglia, including average velocity, steps taken in 30 s, distance travelled in 5 min and step size, were unchanged throughout the post-injection period (*Figure 1—figure supplement 1*). Injection of a second, non-overlapping shRNA against torsinA (AAV-TorsinAshRNA2-GFP) yielded similar results (*Figure 1D*). These data suggest that torsinA KD in the basal ganglia does not result in dystonia.

## Knockdown of torsinA in the cerebellum of adult mice results in dystonia

The cerebellum has also been implicated in DYT1 (*Argyelan et al., 2009*; *Eidelberg et al., 1998*) and there is evidence that it plays a role in other hereditary dystonias (*Calderon et al., 2011*; *Carbon et al., 2013*; *Oblak et al., 2014*). Therefore, we tested whether torsinA KD in the cerebellum of adult rodents produces dystonia. Injection of AAV-torsinA shRNA-GFP resulted in generalized expression of the construct (*Figure 1E*) and significant knockdown of torsinA in the cerebellum (average torsinA levels after KD: 0.2214 ± 0.05) (*Figure 1F*). Mice with acute knockdown of torsinA in the cerebellum developed uncoordinated movements progressing to frequent dystonic postures (*Figure 1G* and *Video 2*). In separate cohorts of mice, injection of a second shRNA (AAV-TorsinA shRNA2-GFP) resulted in comparable symptoms, while injection of AAV-NTshRNA-GFP to the same region had no effect on motor behavior (*Figure 1F*). In patients with dystonia, abnormal co-contraction of agonist and antagonist muscle pairs often underlies abnormal postures (*Frucht, 2013*; *Hughes and McLellan, 1985*; *Yanagisawa and Goto, 1971*). Electromyographic (EMG) recordings in mice with torsinA KD in the cerebellum demonstrated that abnormal posturing of the hind limb in these animals is due to muscle co-contraction, similar to what is seen in patients (*Figure 1H,I*). It is generally postulated that dystonia may be a circuit-level disorder, resulting from the dysfunction of multiple motor-related areas. Therefore, we examined whether concurrent torsinA KD in the cerebellum and basal ganglia would result in more severe symptoms. We found that knockdown of torsinA in the cerebellum and basal ganglia did not enhance the severity of dystonia when compared to mice with torsinA KD in the cerebellum alone (*Figure 1—figure supplement 2*). These findings provide evidence that torsinA KD in the adult cerebellum is sufficient to induce dystonia.

## The severity of symptoms produced by cerebellar knockdown of torsinA is dose dependent

An observation made during the course of our experiments was the variability in the severity of symptoms in animals with cerebellar knockdown of torsinA (*Figure 1G*). To determine the source of the variability, the amount of knockdown of torsinA in each animal was compared to the Dystonia score. It was noted that the severity of dystonic symptoms correlated with the extent of knockdown, such that as knockdown increased so did the severity of symptoms. When torsinA protein level was knocked down by at least 60% dystonic symptoms were observed (*Figure 1J*, p<0.0001). From these data, it is clear that a substantial decrease in torsinA protein levels is necessary to produce dystonia.

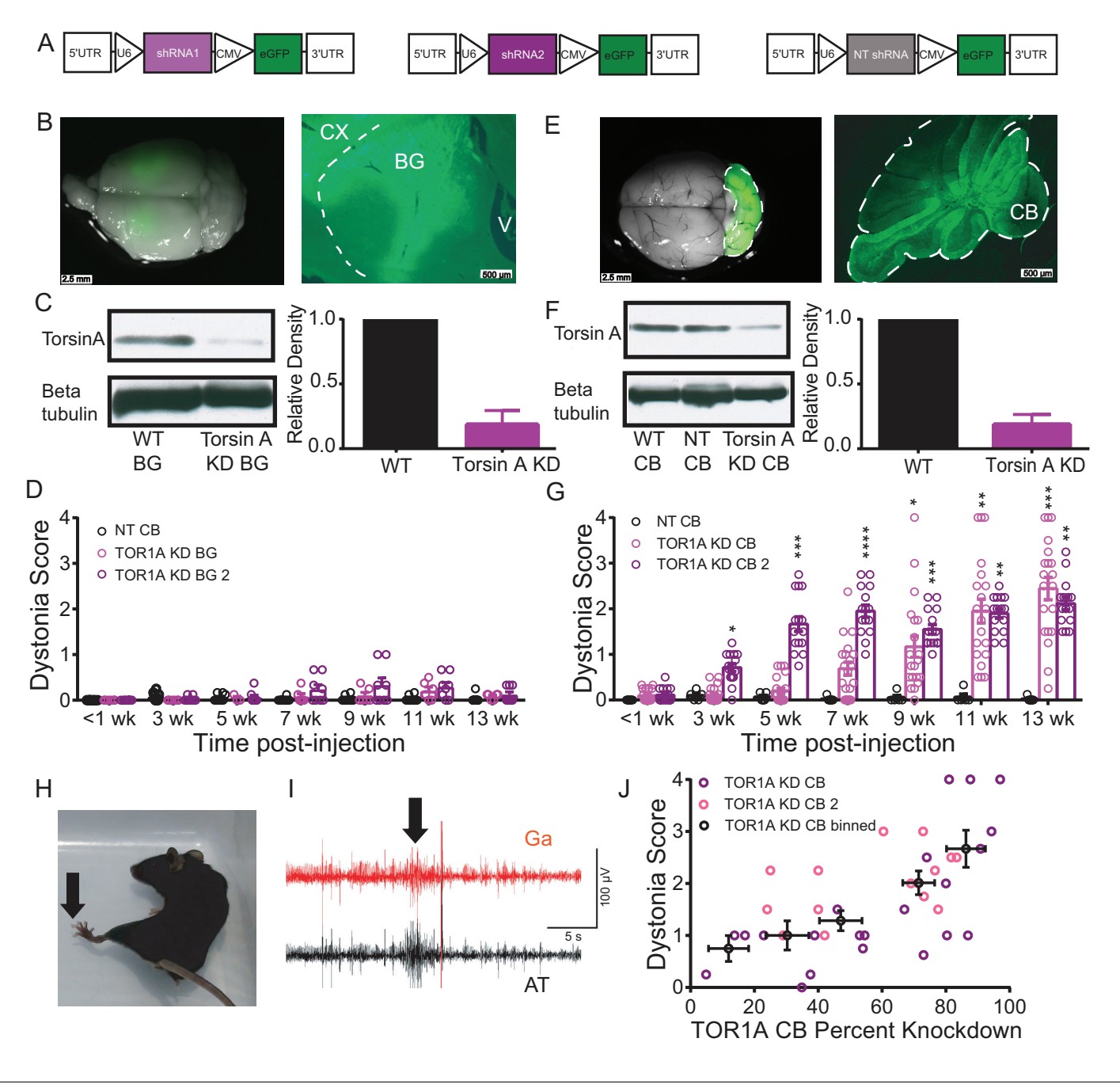

**Figure 1.** Although knockdown of torsinA in the striatum and globus pallidus of adult mice does not cause dystonia, its knockdown in the cerebellum does. (**A**) Schematic of the construct delivered to cells via AAV9 containing either non-targeted shRNA or shRNA against torsinA. Arrows are used to show promoters. (**B**) There was strong GFP expression in the basal ganglia (left image) of mice injected with the shRNA against torsinA. The right image is a sagittal section of a representative mouse injected with AAV-TorsinA shRNA-GFP. The dotted line delineates the separation between basal ganglia and cerebral cortex. CX = cerebral cortex, BG = basal ganglia, V = ventricle. Scale bar = 2.5 mm in left image, 500 μm in right image. (**C**) Example western blot (left) and quantification (right) from three animals in which AAV9 containing shRNA against torsinA was injected into the basal ganglia of adult mice. There is an average knockdown of 81.3% ± 6.2% (Mean ±S.E.M). (**D**) Observers blinded to the condition of the animals rated the basal ganglia injected animals on a previously published dystonia scale. On the graph each point represents an individual animal and the bars represent the average of all animals. All animals up to 13 weeks post-injection had an average score <1 (Mean ± S.E.M, NT shRNA, N = 8; Tor1A KD BG, N = 6; Tor1A KD BG2, N = 8). (**E**) There was strong GFP expression throughout the cerebellum (left image) of mice injected with the shRNA against torsinA. The right image is a sagittal section of a representative mouse injected with AAV-TorsinA shRNA-GFP. The dotted line delineates the cerebellum (CB). Scale bar = 2.5 mm in left image, 500 μm in right image. (**F**) Representative Western blot (left) and quantification (right) showing knockdown of torsinA

*Figure 1 continued on next page*

*Figure 1 continued*

from cerebellar lysates prepared from animals injected with a shRNA against torsinA when compared to WT animals or animals injected with a non-targeted (NT) shRNA. TorsinA KD 77.9% ± 4.7% (Mean ± S.E.M, Tor1A KD N = 6). (**G**) Adult mice with knockdown of torsinA in the cerebellum showed symptoms consistent with dystonia as rated by observers blinded to the condition of the animals. On the dystonia scale, a score of ≥2 is considered dystonic. Adult mice with knockdown of torsinA, using two shRNAs targeted to different regions of the protein, exhibited dystonic symptoms beginning around 7–9 weeks. (*=p<0.05, **=p<0.01, ***=p<0.001, ****=p<0.0001, Mean ± S.E.M, NT shRNA, N = 10; Tor1A KD CB, N = 20; Tor1A KD CB2, N = 15). By 13 weeks post-injection animals on average exhibited symptoms consistent with dystonia on this scale. (**H**) An example mouse 13 weeks after the injection of shRNA against torsinA into the cerebellum exhibiting an abnormal hind limb posture (arrow). (**I**) EMGs performed in three animals confirmed that abnormal postures are due to co-contraction of agonist and antagonist muscle pairs. In this example trace, the gastrocnemius (Ga, red) and anterior tibialis (AT, black) are shown during a dystonic co-contraction (arrow). Scale bar represents 5 s (x) by 100 μV (y). (**J**) The severity of motor symptoms, quantified by the dystonia scale 13 weeks after injection of torsinA in the cerebellum, is correlated with the level of shRNA mediated knockdown of torsinA. As the percent of torsinA knockdown increases, so does the severity of the motor phenotype (Torsin A KD CB (N = 22, purple) and torsin A KD CB2 (N = 15, pink). Black symbols show the average of the binned combined data from both shRNAs against torsinA (Spearman r = 0.63, 95% CI 0.3759 to 0.7960, p value < 0.0001). knockdown is presented as within-blot loading control-normalized values. The data are presented as mean ±S.E.M.

The following figure supplements are available for figure 1:

**Figure supplement 1.** Knockdown of torsinA in the basal ganglia of adult mice does not change open field behavior.

**Figure supplement 2.** Symptoms in adult mice with simultaneous knockdown of torsinA in the basal ganglia and the cerebellum are not significantly different from symptoms seen in mice with knockdown in the cerebellum alone.

This may explain, at least in part, why heterozygous animal models of torsinA dysfunction have routinely lacked overt dystonic symptoms.

## Cerebellar output neurons exhibit erratic burst firing in dystonic animals with knockdown of torsinA in the cerebellum

Previous studies have suggested that abnormal cerebellar output in the rodent can result in dystonia (*Calderon et al., 2011*; *Fremont et al., 2014*, *2015*; *LeDoux et al., 1993*; *Pizoli et al., 2002*). To address whether a similar alteration underlies dystonia in this model, in vivo extracellular recordings were performed in dystonic torsinA KD animals. Recordings from deep cerebellar nuclei (DCN) neurons, which comprise the majority of the cerebellar output, revealed that these cells fired abnormally with bursts of action potentials, whereas DCN cells in controls (NTshRNA) fired tonically (*Figure 2A*). Further, the average firing rate of DCN neurons from dystonic animals was nearly half that of controls (*Figure 2B*, control average firing rate: 55.5 ± 4.1 spikes/second (sp/s), average firing rate in dystonic mice: 22.4 ± 3.4 sp/s, p<0.00001). It has been shown that an increase in the irregularity of cerebellar output can underlie ataxia (*Walter et al., 2006*) and in some cases dystonia in rodents (*Fremont et al., 2014*, *2015*). The regularity of neuronal firing was calculated using the coefficient of variation of the interspike intervals (CV ISI), with increasing CV ISI representing an increase in irregularity. Since the CV ISI is determined by dividing the standard deviation of the interspike interval (ISI) by the mean ISI of the cell, changes in the firing rate will affect the CV ISI. DCN neurons recorded from dystonic animals exhibited a lower firing rate than controls, and would thus be expected to have a lower CV ISI, even without a change in firing regularity. However, the

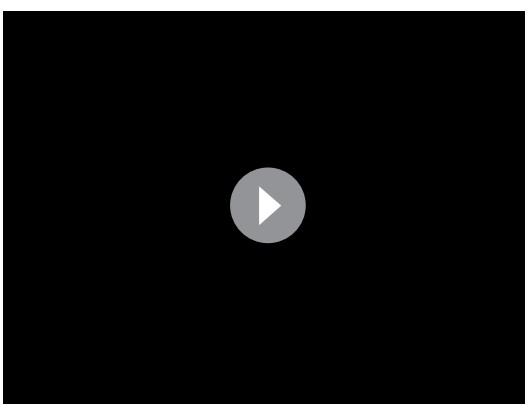

**Video 1.** shRNA-mediated knockdown of torsinA in the basal ganglia of adult mice does not produce dystonic postures. A representative video showing a mouse 13 weeks after injection, injected with a shRNA against torsinA in the cerebellum at 6 weeks of age. In the open-field no motor abnormalities were observed.

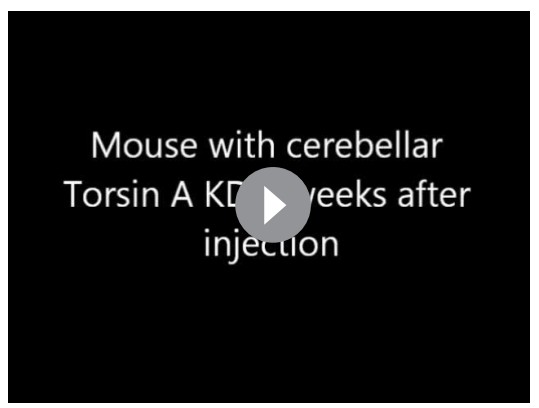

**Video 2.** shRNA-mediated knockdown of torsinA in the cerebellum of adult mice produces dystonic postures by 13 weeks after injection. A representative video showing the progression of symptoms in a mouse injected with a shRNA against torsinA in the cerebellum at 6 weeks of age. The video starts with the mouse walking in an open-field chamber at three weeks after injection displaying no obvious motor abnormalities. At the 24 s time-point in the video, the same mouse is shown at seven weeks after injection with an unsteady gait and jerky movements. At 1 min 10 s, the same mouse is seen at 13 weeks after injection with exacerbated symptoms including difficulty maintaining balance, frequent repetitive movements and dystonic postures.

CV ISI of DCN neurons recorded from dystonic animals was nearly triple that of controls, despite the decrease in average firing rate, suggesting that the activity of these neurons is highly erratic (*Figure 2C*, control CV ISI: 0.55 ± 0.03, dystonic CV ISI: 1.53 ± 0.13, p<0.00001). The CV ISI will increase if the cell begins to fire with long pauses, in high frequency bursts, or both. In a mouse model of the inherited dystonia RDP, high-frequency burst firing was shown to underlie severe dystonia (*Fremont et al., 2014, 2015*). The predominant firing rate (PFR) and the interspike interval (ISI) histogram can identify high frequency firing in dystonic animals. There was a trend towards an increase in the PFR of DCN cells recorded from dystonic mice with torsinA KD compared to controls, (*Figure 2D*, control: 77.3 ± 5.7 sp/s, dystonic animals: 105.1 ± 22.3 sp/s, p=0.21), and there was a peak in the ISI histogram of DCN neurons from dystonic mice at shorter interspike intervals around 2 ms (*Figure 2E*). Taken together, these data suggest that dystonia in mice with torsinA KD is caused by abnormal and erratic cerebellar output.

To determine whether the irregularity of firing changes with the severity of symptoms, awake head-restrained recordings were also performed in mice which developed less severe symptoms not characteristic of dystonia when they were injected with the torsinA shRNA. On the dystonia scale these mice scored between 1 and 2, corresponding to an approximately 30–60% knockdown of torsinA. On average, the changes in the activity of Purkinje cells and DCN neurons in these mice were in between those of the normal and dystonic mice (*Figure 2—figure supplement 1*). However, these data should be interpreted with caution as it is difficult to know whether the data obtained in the less severe mice represents the average behavior of two groups of neurons; namely those with and without knockdown of torsinA, or alternatively it primarily represents less knockdown in the majority of cells. We think the former possibility is the more likely scenario.

## Purkinje cells fire aberrantly in dystonic mice with knockdown of torsinA in the cerebellum

Since DCN activity is strongly influenced by GABAergic input from Purkinje cells, the main computational neurons of the cerebellar cortex, the activity of Purkinje cells was also examined in vivo. Normally, Purkinje cells fire regularly in vivo but in dystonic animals Purkinje cells exhibited abnormal bursting activity (*Figure 2F*). There was no significant difference between the average firing rate of Purkinje cells from dystonic mice (torsinA KD) and those from controls (NTshRNA) (*Figure 2G*, control: 69.1 ± 5.3 sp/s, dystonia: 51.1 ± 12.1 sp/s, p=0.13). However, the CV ISI of Purkinje cells from dystonic animals was increased by over two-fold compared to controls (*Figure 2I*, control: 0.05 ± 0.03, dystonia: 1.21 ± 0.12, p<0.0001). The PFR of Purkinje cells from torsinA KD animals was also significantly increased (*Figure 2H*, control: 85.7 ± 7.1 sp/s, dystonia: 124.6 ± 22.5 sp/s, p=0.031). The ISI histogram of Purkinje cells showed that, similar to the DCN neurons, some cells had short ISIs (around 5 ms compared to a peak ISI at 15 ms for the NTshRNA injected animals, *Figure 2J*). These findings suggest that aberrant Purkinje cell output caused by knockdown of torsinA in the cerebellum could contribute to the abnormal activity of DCN neurons.

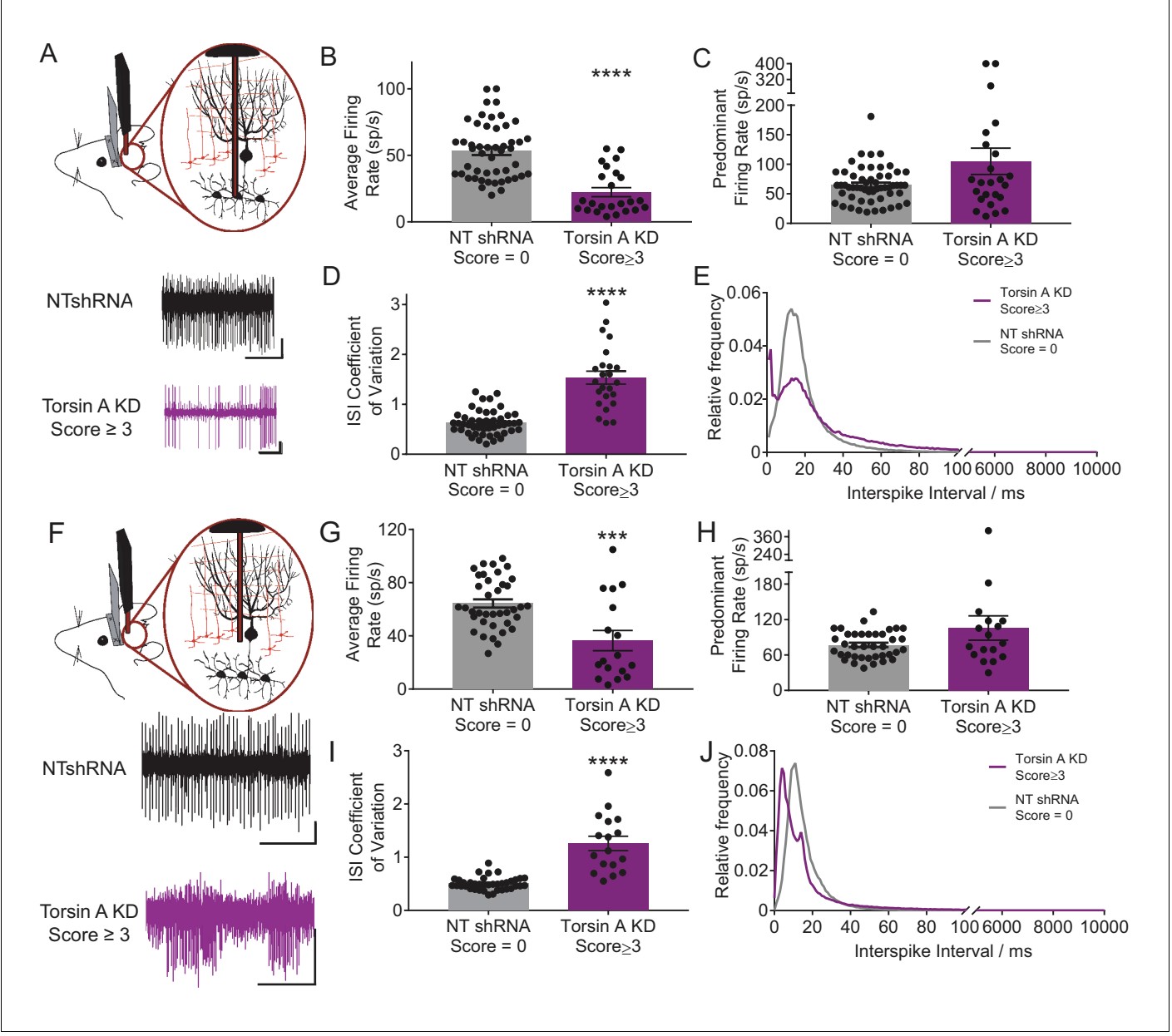

**Figure 2.** Purkinje cells and neurons of the deep cerebellar nuclei (DCN) fire abnormally in dystonic mice. (A) Schematic showing recordings from DCN neurons in awake head restrained animals. To the right are example traces recorded from a non-dystonic mouse injected with non-targeted shRNA (NTshRNA, black) and from a dystonic mouse with cerebellar knockdown of torsinA (torsinA KD, purple). Scale bar = 500 ms (x) by 50 µV (y) for both traces. (B) Quantification of the average firing rate of DCN neurons revealed that DCN cells recorded from dystonic mice with knockdown of torsinA (purple) (N = 3, n = 25) exhibited a significantly decreased average firing rate compared to those from non-dystonic mice expressing NTshRNA (gray) (N = 5, n = 49) (control average firing rate: 53.21 ± 2.998 spikes/second (sp/s), average firing rate in dystonic mice: 22.4 ± 3.4 sp/s, p<0.0001). (D) DCN cells from dystonic mice (purple) also exhibited an increased coefficient of variation of the interspike intervals (CV ISI) compared to controls (gray) indicating that these cells fired more irregularly than controls (control CV ISI: 0.63 ± 0.034, dystonic CV ISI: 1.53 ± 0.13, p<0.0001). (C,E) There was no statistically significant difference between the predominant firing rate of control DCN cells and those from dystonic animals (C) (control: 64.86 ± 4.056 sp/s, dystonic animals: 105.1 ± 22.3 sp/s, p=0.3044), although the interspike interval (ISI) histograms of representative DCN neurons (n = 20 per condition) showed an additional peak at shorter ISIs in dystonic animals (purple, (E) compared to controls (black, (E). On all bar graphs, bars represent mean ± S.E.M with points representing individual values ****= p<0.0001. (F) Schematic showing that Purkinje cells (PCs) were recorded from awake head restrained mice. Beneath are example raw traces of a PC recorded from a control mouse with expression of NTshRNA in the cerebellum (black) and one recorded from a dystonic mouse with knockdown of torsinA in the cerebellum (purple). PCs recorded in dystonic animals exhibited abnormal burst firing. Both scale bars are 200 ms (x) by 200 µV (y). (G) Quantitatively, there was a significant difference between the average firing rate of control PCs (gray; N = 5, n = 38) and those from dystonic mice (purple; N = 3, n = 17) control: 64.31 ± 3.094 sp/s, dystonia: 36.45 ± 7.6 sp/s,

*Figure 2 continued on next page*

*Figure 2 continued*

***p=0.0001. (I) In addition, PCs in dystonic mice (purple) exhibited an increase in the coefficient of variation of interspike intervals compared to controls (gray) suggesting that these cells fire more irregularly (control: 0.5 ± 0.019, dystonia: 1.26 ± 0.135, p<0.0001). (H,J) PCs in dystonic mice (purple) exhibited a trend towards an increased predominant firing rate PFR compared to controls (gray) (control: 77.7 ± 3.76 sp/s, dystonia: 105.6 ± 20.52 sp/s, p=0.24), and the ISI histograms of representative cells from each condition (n = 15 per condition) showed a peak ISI shifted to the left towards shorter ISIs in torsinA KD animals (purple, (J).On all graphs, bars represent mean ± S.E.M with points representing values from individual cells.

The following figure supplement is available for figure 2:

**Figure supplement 1.** Neurons of the deep cerebellar nuclei (DCN) and Purkinje cells recorded in vivo from less severe mice injected with the shRNA against torsinA in the cerebellum exhibit abnormal activity.

Similar to that seen for DCN neurons, Purkinje cells in mice with less severe symptoms also showed abnormal aberrant firing which were, by and large, less altered than those in the dystonic animals (*Figure 2—figure supplement 1*).

## Knockdown of torsinA alters the intrinsic activity of Purkinje cells

Purkinje cells are intrinsically active pacemaking neurons that fire regular action potentials even when deprived of synaptic input (*Batini and Kado, 1967*; *Bell and Grimm, 1969*; *Latham and Paul, 1971*). Since Purkinje cells have high expression of torsinA, alterations in this activity resulting from torsinA KD could contribute to the changes in Purkinje cell firing reported above (*Konakova et al., 2001*; *Puglisi et al., 2013*). To address this possibility, we performed extracellular recordings from Purkinje cells in cerebellar slices in the presence of GABAergic and glutamatergic synaptic blockers. Both Purkinje cells with torsinA KD, identified by their expression of GFP (GFP+ torsinA KD), as well as neighboring neurons without GFP expression (GFP-torsinA KD) were recorded (*Figure 3A*). The Purkinje cells from torsinA KD animals were then compared to those recorded from wild-type animals (WT) and GFP positive (NT GFP+) and negative cells (NT GFP−) from animals injected with AAV9-NTshRNA-GFP. TorsinA KD altered the intrinsic activity of Purkinje cells, converting their pacemaking to irregular burst firing. Although Purkinje cells with torsinA KD had an average firing rate similar to NT and WT (average firing rate of WT: 53.9 ± 6.7, GFP+ TorsinA KD: 68.1 ± 13.1 sp/s, NT GFP+: 49.8 ± 7.0 sp/s, p=0.45, *Figure 3B*), the PFR was nearly doubled (PFR of WT: 53.7 ± 6.6, GFP+ TorsinA KD: 93.2 ± 20.7 sp/s, NT GFP+: 49.5 ± 6.9 sp/s, p=0.048, *Figure 3C*) and the CV ISI was increased by ~10 fold (CV ISI of WT: 0.110 ± 0.010, GFP+ TorsinA KD: 1.151 ± 0.248, NT GFP+: 0.096 ± 0.008, p=0.0003, *Figure 3D*). The ISI histogram also showed that torsinA KD Purkinje cells had a peak ISI shifted to the left similar to the in vivo recordings (*Figure 3E*). GFP− Purkinje cells from dystonic animals had similar firing patterns compared to those from wild-type and AAV9-NTshRNA-GFP controls (*Figure 3B,C,D,E*), providing evidence that torsinA knockdown affects the intrinsic pacemaking of these cells. Together, these findings suggest that knockdown of torsinA disrupts the intrinsic activity of Purkinje cells.

## Knockdown of torsinA alters the intrinsic activity of neurons in the deep cerebellar nuclei

DCN neurons are also intrinsically active, pacemaking cells with high expression of torsinA (*Konakova et al., 2001*; *Puglisi et al., 2013*). In vitro recordings of the intrinsic activity of DCN neurons, with fast glutamatergic and GABAergic synaptic transmission blocked, demonstrated that knockdown of torsinA alters the normal tonic activity of these neurons, causing them to fire at lower rates and with bursts (*Figure 3F*). Extracellular recordings were performed in regions of the DCN with clusters of GFP+ cells (GFP+ DCN) and areas with clusters of GFP− cells (GFP− DCN). Cells were recorded from cerebellar slices of wild type animals, animals injected with AAV9-NTshRNA-GFP (GFP+ and GFP− cells) and animals injected with AAV9-TorsinA shRNA-GFP (GFP+ and GFP− cells *Figure 3G,H,I*). DCN neurons from GFP+ areas of torsinA shRNA injected animals had a mean average firing rate nearly half that of the controls (GFP+ neurons from dystonic animals: 7.9 ± 2.3 sp/s, p=0.0011, GFP+ neurons from control animals: 21 ± 2.1 sp/s, *Figure 3G*). Further, DCN neurons recorded from GFP+ areas in slices from dystonic mice exhibited a ten-fold increase in the CV ISI compared to controls (GFP+ neurons from dystonic animals: 1.21 ± 0.312, GFP+ neurons from

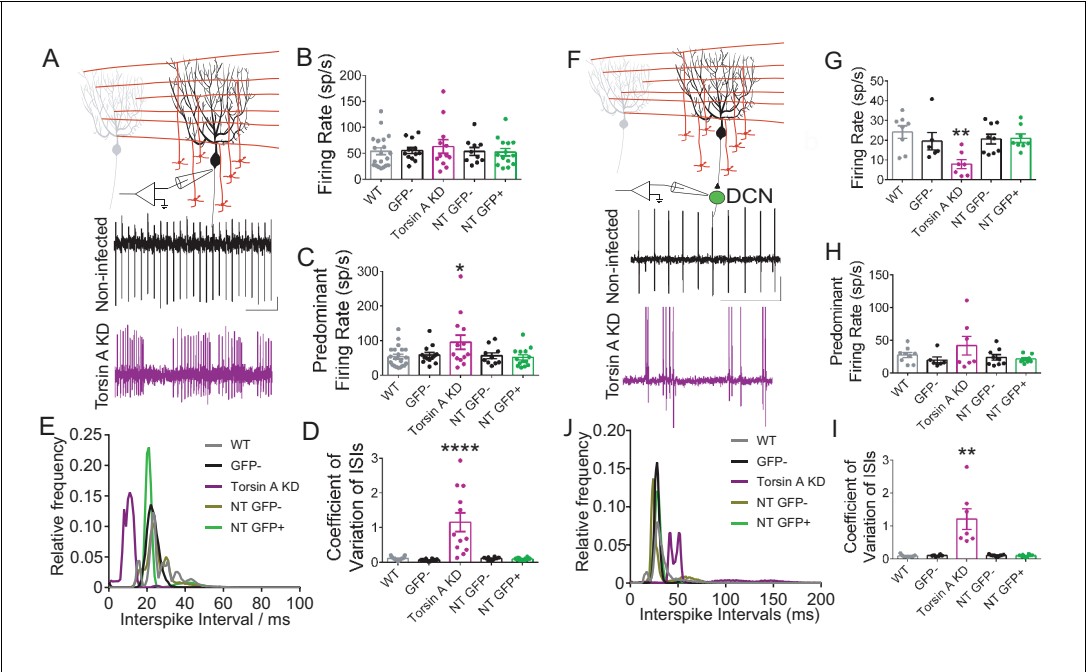

**Figure 3.** Knockdown of torsinA alters the intrinsic activity of Purkinje cells and neurons in the deep cerebellar nuclei (DCN). (A) Schematic of extracellular recordings which were performed from acute brain slices in which fast glutamatergic and GABAergic synaptic transmission was blocked. Beneath are raw traces of adjacent Purkinje cells recorded from dystonic mice with torsinA knockdown in the cerebellum. The Purkinje cell on the top is not infected and exhibits regular firing characteristic of this cell type (black). The one on the bottom had torsinA knockdown and exhibited erratic firing (purple). Scale bar is 200 ms (x) by 50 μV (y) and is applicable to both traces. (B) There is no significant change in the average firing rate of Purkinje cells with torsinA knockdown (TorsinA KD (GFP+ TorsinA KD, N = 5, n = 12) compared to GFP− Purkinje cells (GFP− TorsinA KD, n = 12) from the same animal and Purkinje cells from control animals WT, (N = 5, n = 21) and GFP positive (NT GFP+, N = 3, n = 15) and negative cells (NT GFP−, n = 10) from animals infected with AAV9-NTshRNA-GFP (average firing rate of WT: 53.9 ± 6.7, GFP+ TorsinA KD: 68.1 ± 13.1 sp/s, GFP+ NT: 49.8 ± 7.0 sp/s, p=0.45). (C) Purkinje cells with torsinA knockdown have a higher predominant firing rate than controls (PFR of WT: 53.7 ± 6.6, GFP+ TorsinA KD: 93.2 ± 20.7 sp/s, GFP+ NT: 49.5 ± 6.9 sp/s, p=0.048). (D) Purkinje cells with torsinA knockdown have an increased CV ISI compared to controls reflecting the irregularity of the firing of these neurons (CV ISI of WT: 0.110 ± 0.010, GFP+ TorsinA KD: 1.151 ± 0.248, GFP+ NT: 0.096 ± 0.008, p=0.0003). On all graphs, bars represent mean ± S.E.M with points representing individual values. *=p<0.05, ****= p<0.0001. (E) ISI histogram of all cells in each condition shows that cells with knockdown of torsinA (purple) have shorter ISIs compared to cells from all other conditions. (F) Schematic showing extracellular recordings of DCN neurons in slices where fast glutamatergic and GABAergic synaptic transmission was blocked. Below are representative traces of DCN neurons recorded in slices taken from dystonic mice with knockdown of torsinA. The top trace is a neuron recorded from an area of DCN with no GFP expression which exhibits regular pacemaking characteristic of these cells. The bottom trace is from a neuron located in an area of DCN with high GFP expression and torsinA knockdown which exhibits irregular activity. Scale bar = 500 ms (x) by 200 μV (y) and is applicable to both traces. (G) The average firing rate of DCN cells with torsinA knockdown (torsinA KD N = 3; GFP+ cells n = 7 and GFP− cells n = 6) was significantly reduced compared to DCN cells recorded in control animals (wild type = WT (N = 3, n = 8), control animals injected with AAV-NTshRNA-GFP (N = 3): neurons from GFP+ DCN (NT GFP+ DCN, n = 8) and neurons from GFP− DCN (NT GFP− DCN, n = 9) (average firing rate of GFP+ neurons from dystonic animals: 7.9 ± 2.3 sp/s, GFP+ neurons from control animals: 21 ± 2.1 sp/s **=p<0.01. (H) There was no significant difference between the predominant firing rates (PFR) of DCN neurons with torsinA knockdown compared to controls (GFP+ TorsinA: 41.9 ± 14.2 sp/s, GFP+ NT DCN: 21.4 ± 2.2 sp/s, p=0.15). (I) DCN neurons with torsinA knockdown exhibited an increased coefficient of variation of interspike intervals compared to controls, reflecting the irregularity of firing of these neurons (GFP+ neurons from dystonic animals: 1.21 ± 0.312, GFP+ neurons from control animals: 0.1 ± 0.011, **p=0.0021). (J) ISI histogram of all cells for each condition shows that cells with knockdown of torsinA (purple) have larger ISIs compared to cells from all other conditions. On all graphs, bars represent mean ± S.E.M with points representing individual values.

control animals: 0.1 ± 0.011, p=0.0021, *Figure 3I*). Although there was a trend towards an increased PFR in neurons with torsinA KD, it was not statistically significant (GFP+ TorsinA: 41.9 ± 14.2 sp/s, GFP+ NT DCN: 21.4 ± 2.2 sp/s, p=0.15, *Figure 3H*). The ISI histogram of DCN neurons in GFP+ areas showed that these cells had larger ISIs compared to all other conditions, which is reflected in their decreased average firing rate (*Figure 3J*). These combined in vitro electrophysiological findings suggest that knockdown of torsinA in the adult mouse cerebellum affects the intrinsic activity of at

least two populations of cerebellar neurons, Purkinje cells and neurons of the DCN, likely contributing to dystonia.

## Knockdown of torsinA in the cerebellum results in localized cell death within the neurons of the deep cerebellar nuclei

Even though the function of torsinA in the nervous system remains elusive, it is plausible that the effects of torsinA KD in cerebellar neurons could also result in cell death. Although patients with DYT1 exhibit no gross morphological changes in the brain, there may be more subtle cell loss in certain brain areas (*McNaught et al., 2004*). Further, rodent models of DYT1 suggest that loss of torsinA results in apoptosis and gliosis in particular regions including the cerebellum (*Liang et al., 2014*). To examine whether dystonic mice with torsinA KD in the cerebellum exhibited cell death, TUNEL staining of apoptotic cells was performed post-mortem on the brains of AAV9-TorsinA shRNA-GFP and AAV9-NT shRNA-GFP injected animals (*Figure 4A*). An increase in apoptotic cells was identified in dystonic mice compared to control animals injected with a non-targeted shRNA (*Figure 4A*). Interestingly, even though both Purkinje cells and DCN neurons were found to intrinsically fire abnormally in the dystonic mice, cell death was comparable in the cerebellar cortex of control animals injected with the non-targeting (NT) shRNA and those with torsinA knockdown (Percent cell death in torsinA KD mice: 3.53 ± 0.54 compared to NT shRNA mice: 4.3 ± 0.95) (*Figure 4B*). However, there was noticeable cell death in the DCN of mice with torsinA KD (Percent cell death in

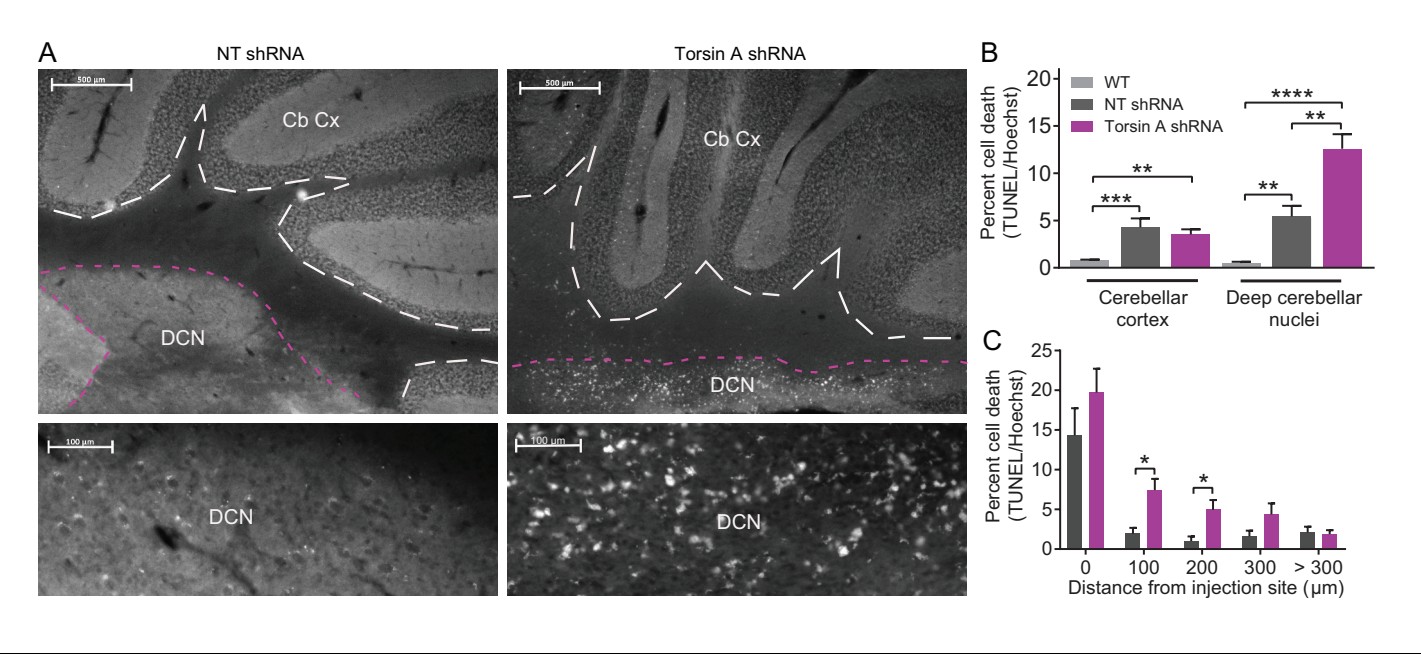

**Figure 4.** Apoptotic cells in the deep cerebellar nuclei (DCN) were observed in mice with torsinA knockdown in the cerebellum. At 15 weeks post-injection, dystonic mice injected with AAV-TorsinAshRNA-GFP (N = 6), age-matched mice injected with AAV-NTshRNA-GFP (N = 6) and wild-type mice (N = 4) were perfused, cryosectioned and stained using a TUNEL assay protocol. (A) Sections were stained for TUNEL and Hoechst. Representative images compare the density of TUNEL positive cells in cerebellar cortex (Cb Cx) and the deep cerebellar nuclei (DCN). Dashed lines outline the Cb Cx and dotted lines outline the DCN. Note the increase in TUNEL positive cells (white) in the DCN of AAV-TorsinA shRNA-GFP injected animals. Bottom panels are magnified images of the DCN. (B) Quantification of percent cell death in the DCN and Cb Cx. There is a significant increase in apoptotic cells in the Cb Cx and DCN of AAV-TorsinA shRNA-GFP injected mice and non-targeted (NT) injected mice. While there was no difference in the cerebellar cortex between the percentage of apoptotic cells in AAV-NTshRNA-GFP mice compared to AAV-TorsinAshRNA-GFP mice, there was a significant increase in apoptotic cells in the DCN of AAV-TorsinAshRNA-GFP compared to AAV-NTshRNA-GFP mice. Quantification of percent cell death in relation to distance from the injection site shows that most of the apoptotic cells from the AAV-NTshRNA-GFP mice were located at the injection site. Mice injected with AAV-TorsinAshRNA-GFP displayed a significant increase in cell death up to 200 μm from the injection site when compared to the AAV-NTshRNA-GFP mice. (*p<0.05, **p<0.01, ***p<0.001, ****p<0.0001 Mean ± SEM) (Scale bar = 500 μm in images in the top panel; scale bar = 100 μm in magnified images of the DCN, bottom panels).

torsinA KD mice: 12.59 ± 1.54 compared to NT shRNA mice: 5.43 ± 1.14). Closer inspection showed that at the injection site both NT- and torsinA- shRNA injected mice had significant cell death compared to age-matched wild-type mice both in the cerebellar cortex and deep cerebellar nuclei. This cell death was most likely caused by the injury associated with the injection as, for both the NT and torsinA injected mice, most of the cell death was seen at the injection site. Nonetheless, significant cell death in the DCN was observed up to 200 μm away from the injection site in the torsinA injected animals but not in the NT injected mice (*Figure 4C*), suggesting that the DCN is sensitive to dysfunction of torsinA and that loss of torsinA can lead to DCN cell death.

## Early postnatal knockdown of torsinA in the cerebellum results in attenuated symptoms

The difference in symptoms reported for mice with embryonic knockout of torsinA in the hind brain (*Liang et al., 2014*) and those with acute knockdown of torsinA in the adult is notable. Mice with conditional knockout of torsinA and mice with a conditional knock-in of torsinA in the cerebellum and midbrain using an En1-cre mouse line were mildly ataxic (*Liang et al., 2014*), whereas mice with torsinA KD in the adult mouse cerebellum developed overt dystonic postures. Interestingly, the motor abnormalities of the En1-cre selective knock-in mice attenuated over time (*Liang et al., 2014*), suggesting a compensatory mechanism. Of particular importance is the continued development of the cerebellum postnatally in humans and rodents. Purkinje cells continue their maturation after birth up to 28 days postnatally and granule cells undergo enormous proliferation during this time (*Hatten and Heintz, 1995*). To determine whether the severity of symptoms in mice depends on the age at which torsinA is lost, a cohort of mice was injected with AAV-TorsinA shRNA-GFP during postnatal cerebellar development 7 days after birth (KD Juveniles) (*Figure 5A*). These animals showed knockdown of torsinA (average torsinA levels after KD: 0.16 ± 0.025) comparable to that seen in mice injected as adults (*Figure 5B—figure supplement 1*). However, unlike mice injected in adulthood, mice injected in the early postnatal period developed only mild motor symptoms and no dystonia, regardless of the level of knockdown (*Figure 5C*, *Video 3*). This finding suggests that mice compensate for knockdown of torsinA during development, and it may explain, at least in part, why the development of dystonic animal models of DYT1 has been challenging. Differences in compensation for knockdown between adult and developing animals has also been shown for other neuronal proteins (*Mallucci et al., 2002*; *Mukherjee et al., 2010*; *Nerbonne et al., 2008*; *Yuan et al., 2005*). It is interesting to note that in patients with DYT1, onset of dystonia usually occurs in childhood/early adolescence, the average age of onset is 12, (*Bressman, 2004*) in which most of the brain development has occurred. Therefore, it is likely that our findings in dystonic mice with acute knockdown of torsinA could be relevant to patients.

## Discussion

By acutely knocking down the protein implicated in DYT1 in adult mice, we generated an animal model that faithfully replicates the symptoms and features of this disorder. We find that knockdown of torsinA gradually produces severe and overt dystonia. Remarkably, the cerebellum and not the basal ganglia is implicated as the main site of dysfunction in DYT1 dystonia. This model underscores the relevance of the cerebellum in dystonias, and offers a powerful platform for examination of the neural, neuronal and molecular substrates of dystonia in DYT1.

## The challenge of identifying the mechanism of disease in hereditary dystonias

Although dystonias are the third most common movement disorder we currently have few, if any, effective therapies (*Balint and Bhatia, 2014*; *Stacy, 2006*). A sizable fraction of dystonias are hereditary and are caused by mutations in a variety of different genes (*Bressman, 2004*; *Charlesworth et al., 2013*; *Klein et al., 1999*; *Müller, 2009*; *Németh, 2002*; *Nygaard et al., 1999*; *Paudel et al., 2012*; *Risch et al., 1995*). While giant strides were made when the implicated genes were identified for a number of different hereditary dystonias, by and large to date these discoveries have not translated into new therapeutics or even more effective management of the symptoms (*Balint and Bhatia, 2014*). Plausibly, two setbacks have slowed down progress; (a) we do not have a clear understanding of the functions of most of the genes identified, and (b) the rodent transgenic

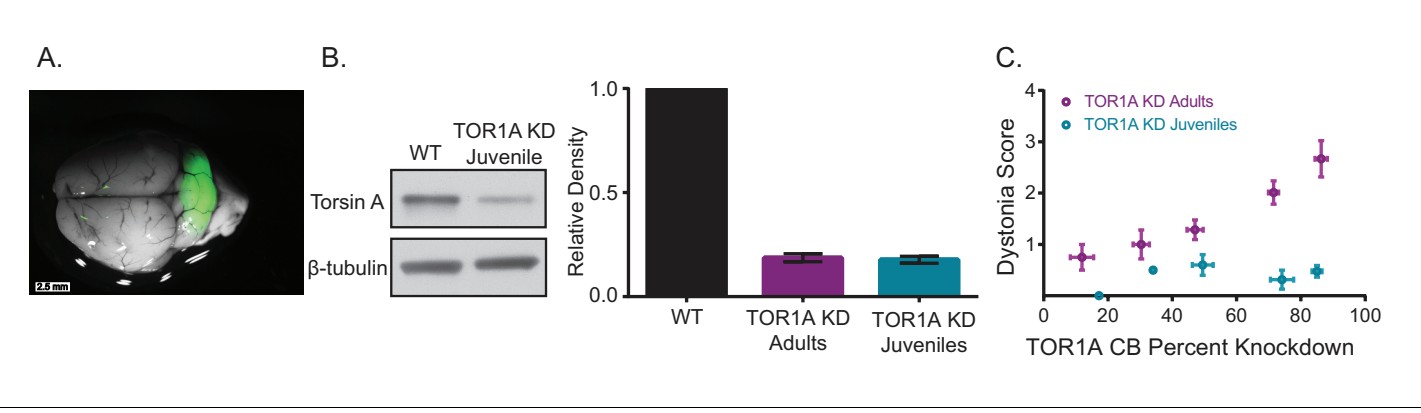

**Figure 5.** Knockdown of torsinA in the cerebellum of mice injected at early postnatal life produces mild motor symptoms not consistent with dystonia. (A) Injection of AAV-TorsinAshRNA-GFP in the cerebellum of mice at postnatal day 7 (P7) results in expression of the construct throughout the cerebellum. (B) Injection of AAV-TorsinAshRNA-GFP in the cerebellum produces similar knockdown in mice injected at P7 (torsin A KD Juveniles) and in mice injected at 6–8 weeks of age (torsinA KD Adults) (Adult KD: N = 17, Juveniles KD: N = 8, Mann-Whitney test, p=0.5294, ns) 13 weeks after injection. The left image displays a representative western blot of cerebellar lysates from a non-injected mouse and a mouse injected at P7. The y-axis in the right graph represents within-blot loading control normalized values. The shRNA against torsinA decreases expression of torsinA in vivo by approximately 80% in both adults and juveniles compared to wild-type mice. (C) shRNA mediated torsinA knockdown in the cerebellum produces dystonia in a dose-dependent manner when mice are injected in adulthood, in contrast to mice injected at early postnatal life that do not develop a severe phenotype independent of the knockdown level. The severity of motor symptoms quantified by the dystonia scale 13 weeks after injection is correlated with the level of shRNA mediated knockdown of torsinA in mice injected at 6–8 weeks of age, such that as the percent of torsinA knockdown increases, so does the severity of the motor phenotype (Spearman r = 0.63, 95% CI 0.3759 to 0.7960, p value < 0.0001). In contrast, mice injected at early postnatal life do not develop dystonia by 13 weeks after injection, independent of the level of torsinA knockdown (Spearman r = 0.1228, 95% CI −0.3803 to 0.5698, p value = 0.6387, ns). The x-axis represents within-blot loading control normalized values. The y-axis represents the average of the dystonia score 13 weeks after injection from four colleagues blinded to the animal's condition. Mice were grouped in bins of 20% knockdown. Each point represents the average of the dystonia score and percent knockdown of the mice within the bin and the lines show SEM.

The following figure supplement is available for figure 5:

**Figure supplement 1.** TorsinA dose response curve.

---

mouse models have not shown overt symptoms, thus making identification of the relevant affected brain regions and pathways difficult. In the case of DYT1, arguably both of these factors have hampered progress.

In most cases, DYT1 is caused by a 3 bp deletion in the torsin1A gene, resulting in the loss of a single glutamate residue in the C-terminus of the protein torsinA (*Ozelius and Lubarr, 1993*; *Ozelius et al., 1997*). Because this mutation causes little, if any, notable neurodegeneration in DYT1, it is thought that dystonia results from dysfunction of neurons (*Konakova et al., 2001*; *McNaught et al., 2004*; *Rostasy et al., 2003*). However, unraveling how the mutation results in neuronal dysfunction has proven to be a challenge. TorsinA is a member of the AAA-ATPase family of proteins and is likely to have a number of different functions in the cell (*Neuwald et al., 1999*; *Vale, 2000*). Because to date the transgenic mouse models of DYT1 have failed to show overt severe dystonia, it has been difficult

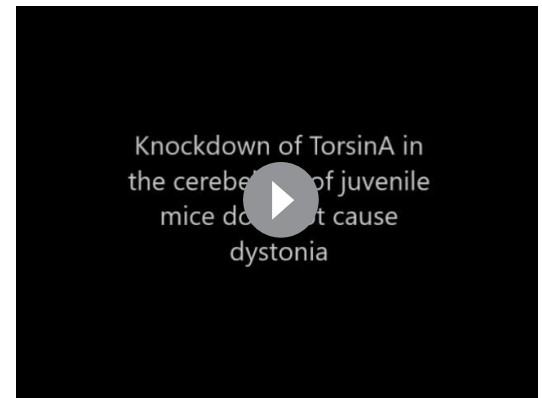

**Video 3.** shRNA-mediated knockdown of torsinA in the cerebellum of juvenile mice does not produce dystonic postures by 13 weeks after injection. A representative video showing a mouse 13 weeks after injection, injected with a shRNA against torsinA in the cerebellum at 1 week of age. In the open-field the mouse displays an unsteady gait but no dystonic postures are observed.

to examine the neural, neuronal, and even molecular substrates of dystonia with certainty (*Dang et al., 2005*; *DeAndrade et al., 2016*; *Goodchild et al., 2005*; *Grundmann et al., 2012*, *2007*; *Page et al., 2010*; *Sharma et al., 2005*; *Shashidharan et al., 2005*). Much research has been carried out on non-symptomatic transgenic mice, and in a few other animal models such as C. elegans and flies (*Beauvais et al., 2016*; *Caldwell et al., 2003*; *Dang et al., 2005*; *Goodchild et al., 2005*; *Muraro and Moffat, 2006*; *Tanabe et al., 2012*). These studies have attributed a number of different functions to the protein, from being a cytoskeletal and nuclear envelope component to serving as protein quality control, contributing to synaptic transmission, and forming part of synaptic vesicle machinery (*Beauvais et al., 2016*; *Caldwell et al., 2003*; *Goodchild et al., 2015*, *2005*; *Grillet et al., 2016*; *Koh et al., 2004*; *Lee et al., 2009*; *Muraro and Moffat, 2006*). However, the absence of faithful symptomatic mouse models has prevented critical evaluation of the importance, or even the relevance, of the various proposed functions of torsinA for dystonia. A good animal model with both genetic and phenotypic validity, therefore, is cardinal for scrutiny of the underlying causes of DYT1, and for design of rational therapies for the disorder.

## Developmental compensation as one of the causes of asymptomatic animal models

It is difficult to know with certainty why most transgenic mouse models of hereditary dystonias have often yielded asymptomatic animals. What is clear, however, is that the failure of the approach is not because rodents do not suffer from dystonia. This assertion is based on the fact that there are a number of spontaneous mutations in hamsters, rats and mice that have resulted in strains that exhibit overt and profound dystonia (*Duchen, 1976*; *Green and Sidman, 1962*; *Jinnah et al., 2005*; *LeDoux, 2011*; *Liu et al., 2015*; *Löscher et al., 1989*; *Wilson and Hess, 2013*). Scrutiny of these models have shown that in some cases dysfunction of the implicated murine genes also generate dystonia, ataxia, or other forms of movement disorders also in humans, thus underlying the potential suitability of mice as a model for examination of dystonia (*Charlesworth et al., 2013*; *Jinnah et al., 2005*; *LeDoux, 2011*). Further, there is strong evidence that other movement disorders with dystonic features can be modelled in rodents. For example, the symptoms of paroxysmal nonkinesigenic dyskinesia was faithfully recapitulated by modifying the causative gene *Pnkd* in mice (*Lee et al., 2012*).

A very plausible explanation for the lack of symptoms in some of the transgenic mouse models of dystonia may be compensation. Genetically, humans are invariably more sophisticated than the lower species used to model disease, and humans often use the same proteins in a far more specialized and dedicated manner. Consequently, the lower species are better poised to compensate for the loss of function of a gene by taking advantage of the increased genetic and signaling plasticity present during embryonic and early postnatal development (*Blendy et al., 1996*; *Chesselet and Carmichael, 2012*; *Dow and Lowe, 2012*; *Pietrobon, 2002*). To underscore this point, it is noteworthy that there are numerous examples where acutely disrupting the function of a protein in mice has major consequences, but knockout of the same gene manifests as few, if any, discernable outcomes (*Daude et al., 2012*; *De Souza et al., 2006*; *Hall et al., 2013*; *Hommel et al., 2003*; *Rossdeutsch et al., 2012*; *Rossi et al., 2015*; *White et al., 2016*). In fact, there are now a number of studies which demonstrate that the effect of knockdown or knockout of a particular protein in the brain can be dependent on when that protein is lost (*Erdmann et al., 2007*; *Mallucci et al., 2002*; *Mukherjee et al., 2010*; *Nerbonne et al., 2008*; *Wang et al., 2003*; *Yuan et al., 2005*).

## knockdown of protein function in adult mice: an alternative approach for generation of rodent models of hereditary dystonias

One approach to overcome compensation is to prevent or reduce the engagement of the potential compensatory mechanisms in mice by acutely targeting the causative gene in the mature animal. Such an approach was successfully employed to generate a faithful animal model of RDP, a movement disorder caused by loss of function mutations in the Na/K pump (*Calderon et al., 2011*; *Fremont et al., 2014*, *2015*). In RDP, subjects carrying the mutation often lead a relatively healthy life until they are exposed to a very stressful event, at which point they rapidly develop severe dystonia and Parkinsonism-like symptoms (*Brashear et al., 2007*, *1996*). Similar to other mouse models of hereditary dystonias, transgenic mice harboring loss of function mutations in the Na/K pump have

generally failed to exhibit overt dystonia (*Clapcote et al., 2009*; *DeAndrade et al., 2011*; *Moseley et al., 2007*). However, in contrast to prior dystonia-related proteins, the function of the Na/K pump as an ion transporter is well-understood and the availability of an exquisitely selective blocker allowed Calderon *et al*. to generate a pharmacologic mouse model of RDP which closely paralleled the disorder in humans (*Calderon et al., 2011*). While partially blocking the function of the Na/K pump in the basal ganglia and the cerebellum of adult mice resulted in mild motor dysfunction, stressing these mice precipitated severe dystonia and Parkinsonism-like symptoms that persisted thereafter (*Calderon et al., 2011*). Further scrutiny of this mouse model surprisingly revealed that dystonia was instigated by the dysfunction of the cerebellum, whereas the Parkinsonism-like features were of basal ganglia origin (*Calderon et al., 2011*; *Fremont et al., 2014*). The success of the pharmacologic model of RDP in replicating the human disorder was recapitulated in vivo in adult mice using short hairpin RNAs to reduce the expression of the Na/K pump in select brain regions (*Fremont et al., 2015*). Reassuringly, the shRNA approach fully corroborated the findings of the pharmacologic model, underscoring a potentially causal role for the cerebellum in generation of dystonia in RDP. The relevance of this finding to the human disorder has been strengthened by the fact that recent postmortem histological studies have identified cerebellar degeneration in RDP (*Sweadner et al., 2016*). Moreover, it has also been recently reported that in some patients mutations in the Na/K pump can result in Adult Rapid Onset Ataxia rather than dystonia (*Sweadner et al., 2016*), a finding that highlights the role of the cerebellum in the disorder and further validates the pharmacologic and shRNA mouse models.

## A dystonic shRNA-based mouse model of DYT1

The success of the acute knockdown models of RDP in replicating the relevant features of the human disorder prompted us to generate a comparable rodent model of DYT1 to explore the possibility that compensation might have hindered efforts in generating faithful transgenic mouse models of DYT1 (*Dang et al., 2005*; *DeAndrade et al., 2016*; *Goodchild et al., 2005*; *Grundmann et al., 2012*, *2007*; *Page et al., 2010*; *Sharma et al., 2005*; *Shashidharan et al., 2005*). Differential compensation in mice compared to humans might be a particularly significant problem in the case of torsinA. This is because expression studies demonstrate that the level of torsinA transcript and protein in neural tissue is markedly different between the developing mouse and human brain (*Siegert et al., 2005*; *Vasudevan et al., 2006*; *Xiao et al., 2004*). In humans torsinA protein is not detectable in neurons until one month after birth when expression abruptly increases to levels maintained throughout adulthood (*Siegert et al., 2005*). In contrast, in the rodent brain torsinA is present at high levels embryonically and peaks during the early postnatal period, decreasing rapidly within a couple of weeks postnatally before stabilizing (*Vasudevan et al., 2006*). Based on this expression pattern it could be argued that if torsinA is necessary for the development of CNS in the mouse but not the human brain, then the mouse brain might more aggressively engage compensatory mechanisms to overcome loss of torsinA during this crucial period than might the human brain. Indeed, the most symptomatic transgenic mouse models to date exhibiting mild abnormal twisting movements were generated using Nestin-cre mice (*Liang et al., 2014*). Even though recombination begins at E11 in Nestin-cre mice, it increases gradually until early postnatal life, which temporally delays neuronal torsinA dysfunction during the gestation period (*Liang et al., 2012*).

To bypass a potentially powerful developmental compensation in mice, we knocked down torsinA in six weeks old adult animals, at a time when cerebellar and basal ganglia development is complete (*White and Sillitoe, 2013*). In agreement with the hypothesis that developmental compensation in mice had prevented generation of symptomatic transgenic mice, we found that acutely knocking down torsinA in the cerebellum of adult mice resulted in overt and severe dystonia. As evidence that compensation plays a major role in the lack of overt symptoms observed in most transgenic mouse models of DYT1, we found that comparably knocking down torsinA in juvenile mice at postnatal day 7, before cerebellar development is completed, did not produce dystonia.

It is important to note that no animal model of a human disorder is ever without a flaw. While torsinA loss of function is present throughout development in human patients, the model described here acutely knocks down torsinA during adulthood, which clearly does not faithfully replicate the human disease. Moreover, it is also unknown whether in humans, a developmental disruption in basal ganglia function contributes to dystonia. However, given that in humans torsinA is not detectable in the brain until four weeks postnatal (*Vasudevan et al., 2006*; *Siegert et al., 2005*), the

shRNA approach described might be a more appropriate mouse model of what occurs in humans than the models that reduce expression of torsinA developmentally when it is expressed at relatively high levels in the rodent brain.

## The potential compensatory mechanisms in transgenic mouse models of DYT1

It is difficult to know what the potential developmental compensatory mechanisms are by which the rodent brain effectively tackles loss of torsinA. While it is possible that compensation engages a number of distinct and parallel processes, the most parsimonious possibility might be the presence of functional redundancy in the family of torsin proteins. In addition to torsinA, there are three other members of the torsin family (*Ozelius et al., 1999*). One of these proteins, torsinB, is a related homologue to torsinA with 70% homology (*Ozelius et al., 1999*).

Both torsinA and torsinB proteins are localized to the endoplasmic reticulum and nuclear envelope membrane system in cells throughout the brain and non-neural cells (*Naismith et al., 2004*). The first evidence of functional redundancy for torsinA was noted in studies performed in DYT1 animals. In DYT1 KI and KO mice, nuclear envelope budding was observed only in neurons in the CNS and no nuclear envelope budding was found in non-neuronal cells or in non-CNS tissue even though torsinA is expressed in all these tissues (*Goodchild et al., 2005*). However, when torsinB was knocked down in fibroblasts from a DYT1 mutant background, nuclear envelope defects were seen in non-neuronal cells as well, demonstrating that torsinB can compensate for loss of torsinA (*Kim et al., 2010*). To determine whether a similar compensation can occur in the CNS, Tanabe et al used embryoid bodies derived from DYT1 KI animals and showed that either torsinA or torsinB can individually rescue the abnormal nuclear envelope budding when the cells are differentiated into neural lineage (*Tanabe et al., 2016*). Interestingly, torsin2A also appeared to rescue nuclear envelope budding to some extent (*Tanabe et al., 2016*). Whether these redundancies can compensate for all the various functions assigned to torsinA, or whether the ability to compensate is shared amongst human torsin proteins is unknown. The presence of such forms of redundancies in rodents may explain, at least in part, the difficulty in recapitulating the human phenotypic symptoms of DYT1 in mice. It is also noteworthy that expression of torsinB like torsinA, differs temporally between mice and humans (*Siegert et al., 2005*; *Tanabe et al., 2016*; *Vasudevan et al., 2006*) adding yet another level of complication to modeling DYT1 in rodents.

If compensation is indeed the mechanism by which the mouse brain can developmentally overcome the shortcomings associated with loss of torsinA function, it suggests that there might be a potentially powerful therapeutic opportunity – understanding the compensatory mechanisms in mice might shed light on strategies for management of torsinA dysfunction in humans. It is tempting to speculate that perhaps some of the same compensatory mechanisms, alas not as effectively or on every occasion, might be engaged in humans and might account for the partial (30–40%) penetrance of DYT1 in humans (*Argyelan et al., 2009*; *Eidelberg et al., 1998*; *Martino et al., 2013*; *Ozelius and Lubarr, 1993*).

## Cerebellum as the instigator of dystonia in DYT1

For a number of years neuroimaging studies in patients with DYT1 have hinted that the cerebellum is involved in dystonia (*Carbon et al., 2010b*, *2008a*, *2008b*; *Eidelberg et al., 1998*; *Trost et al., 2002*; *Zoons et al., 2011*). However, to date it has been difficult to examine the role of the cerebellum in DYT1 dystonia using animal models because region specific transgenic mice have generally failed to recapitulate overt dystonic symptoms (*Liang et al., 2014*; *Pappas et al., 2015*; *Weisheit and Dauer, 2015*; *Yokoi et al., 2011*, *2008*; *Zhang et al., 2011*), as discussed perhaps likely because of developmental compensation. Our approach allowed for targeted knockdown of torsinA in select brain regions including the cerebellum, and provided strong evidence that the cerebellum may be causative for instigating dystonia in DYT1, thus supporting the findings of the neuroimaging studies done in patients.

Based on patient studies, the notion that the cerebellum might be involved in some dystonias has been around for decades and a number of reports have unequivocally documented that the cerebellum is abnormal in a number of different dystonias (*Alarcón et al., 2001*; *Carbon et al., 2010a*; *Filip et al., 2013*; *Jinnah and Hess, 2006*; *Neychev et al., 2008*; *Prudente et al., 2014*;

*Simonati et al., 1997*; *Trost et al., 2002*). However, human studies are limited in their scope and have not been in a position to determine whether the cerebellum causally contributes to dystonia, or whether the noted abnormalities are compensatory in nature. Given the diversity of dystonias, it would be surprising if both cases are not rampant. Nonetheless there are clear examples of dystonia in patients that appear to be causally related to cerebellar dysfunction, particularly when dystonia manifests as a symptom in cerebellar-centric disorders such as spinocerebellar or episodic ataxias (*Bang et al., 2003*; *Kawarai et al., 2016*; *Mariotti et al., 2007*; *Muglan et al., 2016*; *Nakagaki et al., 2002*; *Sethi and Jankovic, 2002*). In some cases, lesioning or stimulation of the cerebellum has been documented to improve the symptoms, corroborating the notion that in some patients, dystonia can arise from cerebellar dysfunction (*Koch et al., 2014*; *Panov et al., 2013*; *Teixeira et al., 2015*).

Our dystonic mouse model of DYT1 allowed us to explore how cerebellar dysfunction causes dystonia. We found, similar to what was noted some twenty years ago in the dystonic rat and more recently in dystonic mouse models of RDP, that cerebellar-induced dystonia is associated with bursting cerebellar output (*Fremont et al., 2014*, *2015*; *LeDoux and Lorden, 1998*). Our data along with a number of other mouse models of dystonia suggest that a bursting cerebellar output may be a hallmark of cerebellar-induced dystonia (*Fremont et al., 2014*; *LeDoux et al., 1998*; *LeDoux and Lorden, 1998*; *Lorden et al., 1992*). Notably, similar bursting cerebellar output has also been documented in a dystonic patient undergoing surgery (*Slaughter et al., 1970*). It is important to note that in the case of cerebellar-induced dystonia animal studies suggest that silencing, lesioning or otherwise normalizing cerebellar output would be an effective therapeutic approach (*LeDoux et al., 1993*; *Neychev et al., 2008*), and recent reports document that cerebellar deep brain stimulation and transcranial magnetic stimulation might also be effective in lessening dystonia in patients (*Koch et al., 2014*; *Teixeira et al., 2015*).

It is intriguing that cerebellar-induced dystonia is seemingly indistinguishable from dystonia caused by the dysfunction of the basal ganglia. A plausible explanation might be that the aberrant cerebellar output ultimately disrupts basal ganglia function to the extent that the disorder manifests as if the dysfunction had originally arisen from the basal ganglia. Scrutiny of the dystonic mouse model of RDP suggests that the disynaptic projection from the cerebellum to the dorsolateral basal ganglia via the thalamus, which is present both in rodents and nonhuman primates, might be the conduit by which the abnormal cerebellar output disrupts basal ganglia function to cause dystonia (*Chen et al., 2014*). Indeed, severing the link between the cerebellum and basal ganglia by either silencing the relevant thalamic neurons, or lesioning the appropriate region can effectively alleviate cerebellar-induced dystonia in mice (*Calderon et al., 2011*; *Chen et al., 2014*).

With regards to DYT1, what remains to be established is how loss of torsinA function disrupts cerebellar activity. Recently, PNKD, the protein disrupted in paroxysmal nonkinesigenic dyskinesia was found to play a crucial role in neurotransmitter release by regulating exocytosis (*Shen et al., 2015*). Several roles for torsinA have been suggested by a number of studies including potential changes in synaptic transmission, endoplasmic reticulum (ER) stress, and abnormal levels of eukaryotic initiation factor 2α (eIF2α) in the dystonic phenotype in DYT1 (*Beauvais et al., 2016*; *Rittiner et al., 2016*). Our data provide little mechanistic insight as to the cause of cerebellar dysfunction, although it is clear that at the very least the intrinsic activity of both Purkinje cells and DCN neurons are altered by loss of torsinA. The change in intrinsic activity could be due to a direct effect of loss of torsinA, or indirectly through alterations in synaptic transmission. The animal model described here provides a valuable platform for future mechanistic studies aimed at elucidating the cellular and molecular pathways that contribute to dystonia in DYT1.

To date, the neural substrates of dystonia have been unclear with studies in patients implicating abnormal activity in multiple motor-related areas and circuits (*Neychev et al., 2011*; *Zoons et al., 2011*). While our findings are not the first to suggest a role for the cerebellum in dystonia (*LeDoux et al., 1993*; *Pizoli et al., 2002*), we provide compelling evidence that abnormal cerebellar output may play a role in DYT1, the most common inherited dystonia. Further, the model we have generated faithfully recapitulates the most common and debilitating symptom in DYT1, dystonia and therefore provides a platform for gaining further understanding into this disorder and developing novel therapeutics.

# Materials and methods

Experiments were performed on 8 to 10 weeks old male or female C57BL/6 mice. All experiments were performed in accord with the guidelines set by Albert Einstein College of Medicine, under the senior investogator's Institutional Animal Care and Use Committee (IACUC) approved protocol 'Cerebellar Function in Health and Disease'.

## shRNA sequences

Two different shRNAs against unique aspects of the sequence of each protein examined were identified. The sequences were originally generated by the RNAi consortium. For torsinA, the identified sequences correspond to TRCN0000008485 (5'-CCGGCCGGAACCTCATAGATTATTTCTCGAGAAA TAATCTATGAGGTTCCGGTTTTT-3') and TRCN0000008487 (5'-CCGGGCTGCAGAAAGATCTGGA TAACTCGAGTTATCCAGATCTTTCTGCAGCTTTTT-3'). Lentiviruses containing these shRNAs were generated by the Albert Einstein College of Medicine Lentiviral Core, and the efficacy of knockdown was tested in vitro in mouse cortical cultures (data not shown, see below and above). Albert Einstein College of Medicine Lentiviral Core generated plasmids containing the effective shRNA sequences. AAV9 compatible plasmids containing each shRNA were produced commercially (Virovek, AAV9-U6-shRNA-CMV-GFP, AAV9-U6-shRNA2-CMV-GFP, AAV9-U6-shRNA119308-CMV-GFP (Lot# 12–183)). The same company also generated AAV9 virus with an average titer of $2 \times 10^{13}$ vg/ml. A 0.22 µm filter sterilized solution containing the virus in DPBS buffer with 0.001% pluronic F-68 was directly injected into different brain regions of adult mice. Control AAV9 containing non-targeted (NT) shRNAs under the same promoter containing the same CMV driven GFP and at an equivalent titer were purchased from Virovek as was AAV9-CMV-GFP. The NT-shRNA used were: AAV9-U6-Non-targeted-Control-CMV-GFP ((5'-GAGGATCAAATTGATAGTAAACCGTTTTGGCCACTGACTGACGG TTTACTATCAATTTGATCCTCTTTTT-3') Lot# 13–234, Virovek) and AAV9-U6-Non-targeted-Control-CMV-GFP ((5'-CCAACTACCCGAACTATTATTCAAGAGATAATAGTTCGGGTAGTTGGCATTTTTT-3') Lot# 13–037, Virovek)

## Cortical cultures and infection with shRNAs

Before injection into live animals, all shRNAs were tested in vitro to determine a relative knockdown efficiency. Cortical neuron cultures were prepared from embryonic day 16 (E16) C57BL/6 mice (Charles Rivers). The pregnant mouse was anesthetized using isoflurane (5%) and the embryos were dissected out and transferred to ice-cold HEPES-Glucose buffer (HEPES 10 mM (Fisher), Glucose 33 mM (Fisher), in PBS (Corning)). Embryos were decapitated and cortices were dissected from the rest of the brain and incubated in HEPES-Glucose buffer containing Trypsin at 37°C for 15 min. Trypsin was washed with HEPES-Glucose buffer then replaced by DMEM (Dulbecco's Modified Eagle Medium, GIBCO) fortified with 10% FBS (Fetal Bovine Serum, GIBCO). Cortical tissues were fragmented by pipetting with a Pasteur pipette heated to obtain a narrow bore. The cells were plated in 24-well culture plates at a density of 20,000 per $cm^2$. Cultures were incubated at 37°C for 2 hr. The culture media was replaced with Neurobasal medium (GIBCO) fortified with B27 supplement (2%, GIBCO) and Glutamax (0.25%, GIBCO). After 5 days, the cultures were transfected with adeno-associated viruses (AAV) carrying shRNA against the target protein's mRNA. Cultures were then incubated for 10 days at 37°C. Neuronal lysates were subsequently collected using lysis buffer (SDS 2%, Tris 50 mM, EDTA 2 mM).

## Western blotting

Protein lysates were prepared and homogenized in an SDS-based lysis buffer containing 2% SDS, 50 mM Tris and 2 mM EDTA (pH = 8). Brain lysates were prepared on ice. Lysates from cultures were prepared at room temperature. All lysates were incubated at room temperature for 30 min for protein digestion, and then sonicated and frozen at −20°C. The protein concentration was determined by BCA assay (Pierce). Samples were run on SDS page gel (BioRad) and transferred to PVDF membrane (BioRad). Membranes were blocked with milk and subsequently incubated with primary TorsinA antibody for 2 hr or less at room temperature on a rocker followed by the secondary antibody for 30 min. PVDF membranes were developed using chemiluminescence and signals were quantified using ImageJ to perform densitometry. The following primary antibodies were used: torsinA (Abcam 34540, 1:1000), beta-tubulin (Sigma T5201, 1:10,000), GAPDH (Cell Signaling 2118 1:5000), B-actin

(Sigma A1978 1:20,000). Beta-tubulin was used as a loading control in all studies because it is an abundant housekeeping protein.

## Stereotaxic injection of shRNAs

All injections were performed in a single surgery. During the same surgery, EMG wires and/or a bracket for in vivo recordings were also implanted.

## Cerebellum

AAV9-shRNA-GFP was injected in mice at 4 sites of the cerebellum. At a rate between 0.1 and 0.15 µl/min, a total volume of two microliters was injected at each site. After the injection, the syringe was left in place for a minimum of 10 min before it was removed. The coordinates used were: (AP:−6 mm, ML:0 mm, DV:−1.5 mm), (AP:−6.96 mm, ML:0 mm, DV: 1.5 mm), (AP: −6 mm, ML: 1.8 mm, DV: 2.3 mm), (AP: −6 mm, −1.8 mm, DV: 2.3 mm). Striatum and globus pallidus: Injections were performed at four sites with two microliters injected at each site at rates identical to cerebellar injections. The coordinates used were: (AP: 0.5 mm, ML: 2 mm, DV: 2.5 mm), (AP: 0.5 mm, ML: −2 mm, DV: 2.5 mm), (AP: −0.5 mm, ML: 2.5 mm, DV: 3.5 mm), (AP: −0.5 mm, ML: −2.5 mm, DV: 3.5 mm). Injections into the substantia nigra were performed at 4 sites with 0.5–1 µl infused at each site at a rate of 0.05 µl per minute. The injections coordinates were: (AP: −3 mm, ML: 1.5 mm, DV: 4 mm), (AP: −3 mm, ML: −1.5 mm, DV: 4 mm), (AP: −3.6 mm, ML: 1 mm, DV: 4 mm), (AP: −3.6 mm, ML: −1 mm, DV: 4 mm).

## Immunohistochemistry

Mice were trans-cardially perfused with 1X PBS followed by 4% paraformaldehyde. Brains were then extracted, kept at four degrees overnight in 4% paraformaldehyde and switched into a 30% sucrose solution before embedding them in Tissue-Tek OCT compound. Brains were sectioned at a 50 µm thickness. Sections were either Nissl stained or stained with an anti-GFP antibody (primary: 1:500, A11122, molecular probes; secondary: 1:1000, A11008, molecular probes) and Hoechst 3342 (1:1000, H3570, molecular probes). Images were taken on a Zeiss Axioskop two plus.

## Dystonia scale

The presence of dystonia and its severity were quantified using a previously published scale (*Calderon et al., 2011*). Briefly, 0 = normal behavior; 1 = abnormal motor behavior, no dystonic postures; 2 = mild motor impairment, dystonic-like postures when disturbed; 3 = moderate impairment, frequent spontaneous dystonic postures; 4 = severe impairment, sustained dystonic postures. The videos were assessed independently by four observers who were blinded to the animal's condition. The observers were trained with a video set containing representative examples for each score in which key characteristics were highlighted.

## EMGs of agonist-antagonist muscle pairs

Electrodes were inserted into the gastrocnemius muscle (extensor) and anterior tibialis muscle (flexor) of the same leg and wires were routed under the skin to a connector fixed to the skull on animals with a dystonia score greater than 2. Immediately prior to recordings, a head-stage for the Pinnacle EEG/EMG recording system 4100 was secured to the connector. For 1–10 min, recordings of muscle activity were performed in the open field while mice were simultaneously videotaped. Afterward, the connector was removed and the animal was placed back into the home cage.

## In vitro electrophysiology

Adult mice were anesthetized with isoflurane and decapitated. Sagittal slices of 300 µm thickness were cut from the cerebellum at a temperature of 35°C and then transferred to room temperature after 1 hr. Extracellular recordings were obtained from single Purkinje cells using a custom-built amplifier and glass pipette electrodes filled with extracellular solution (in mM NaCl 125, KCl 2.5, NaHCO$_3$ 26, NaH$_2$PO$_4$ 1.25, MgCl$_2$ 1, CaCl$_2$ 2, glucose 10, pH 7.4 when gassed with 5%CO2:95% O2). Slices were recorded on an upright microscope (Zeiss) and perfused with 35°C extracellular solution at a rate of 1.5 ml/min during recording sessions. The perfusion solution contained picrotoxin (10 uM) and CGP55845 (1 µM) to block inhibitory synaptic transmission and kynurenic acid (5

mM) to block excitatory transmission. Data was sampled at 10 kHz using an analog to digital converter (National Instruments SCB-68) and analyzed using custom-written LabView software (available upon request).

## In vivo electrophysiology

Mice were implanted with an L-shaped bracket that was fixed onto the skull with bone screws and dental cement. A recording chamber was drilled in the skull on top of the cerebellum, surrounded with dental cement and covered with surgi-foam and bone wax. Single-unit neural activity was recorded extracellularly using a tungsten electrode (Thomas Recording, 2–3 MΩ), which was advanced into the cerebellum until either the Purkinje cell layer or the deep cerebellar nuclei were reached. Purkinje cells were identified by location, characteristic firing rate, the presence of complex spikes, and post-hoc histology. DCN neurons were identified by location, firing rate, and post-hoc histology. Signals were filtered (200 Hz–20 kHz) and amplified (20000x) on a custom built amplifier and then digitized (20 kHz) using a National Instruments BNC-2110. Waveforms were sorted offline using characteristics of the spikes such as amplitude and energy, as well as those determined by principal component analysis (Plexon).

## Cell death quantification

TorsinA injected animals began exhibiting dystonic postures ~13 weeks after injection. At 15 weeks post-injection AAV9-TorsinA shRNA-GFP and AAV9-NT shRNA-GFP animals were trans-cardially perfused with 1X PBS followed by 4% paraformaldehyde (PFA). Brains were removed, incubated at 4°C overnight in 4% PFA and then transferred to a 30% sucrose solution. Brains were then cryopreserved in an OCT compound (Tissue-Tek). The cerebellum was cryosectioned coronally in 30 μm slices using a cryostat maintained at −20°C. Slides were imaged using a Zeiss Axioskop two microscope to identify the injection sites. Slides containing sections up to 1 mm lateral to the injection site were stained using the TUNEL protocol (Roche) to label cells undergoing apoptosis. Slides were also co-stained with Hoechst to quantify the total number of cells. The number of TUNEL positive (red) and Hoechst positive cells (blue) were quantified in 300 × 300 μm sections imaged at different distances from the injection sites. The ratio of apoptotic cells/total number of cells was determined and then used to compare cell loss in the cerebellar cortex and DCN. Images were acquired from up to 1 mm lateral, anterior, posterior, and ventral from all four injection sites in each animal.

## Statistics

For comparing characteristics of firing a student's t-test was used unless otherwise stated. For analysis of behavioral data on the dystonia scale, a Friedman test with post-hoc Dunn's multiple comparison's test or a Mann-Whitney test were used to determine significance. In all cases data was considered statistically different from control when $p < 0.05$. Sample size was not determined using a priori power analysis, but was based on the statistical criteria for significance in observations. Experiments were repeated in at least five mice, and the number of samples are noted in text and the appropriate figure legend.

## Additional information

### Funding

| Funder | Grant reference number | Author |
| --- | --- | --- |
| National Institute of Neurological Disorders and Stroke | R01NS079750 | Kamran Khodakhah |
| National Institute of Neurological Disorders and Stroke | RO1NS050808 | Kamran Khodakhah |
| National Institute of Neurological Disorders and Stroke | F30NS071665- | Rachel Fremont |

The funders had no role in study design, data collection and interpretation, or the decision to submit the work for publication.

## Author contributions
RF, Conceptualization, Data curation, Formal analysis, Funding acquisition, Validation, Investigation, Visualization, Methodology, Writing—original draft; AT, Data curation, Formal analysis, Validation, Investigation, Visualization, Methodology, Writing—original draft, Writing—review and editing; CA, Data curation, Formal analysis, Investigation, Writing—original draft; KK, Conceptualization, Data curation, Supervision, Funding acquisition, Investigation, Project administration, Writing—review and editing

## Author ORCIDs
Rachel Fremont, http://orcid.org/0000-0001-5285-7899
Ambika Tewari, http://orcid.org/0000-0002-8740-5080
Chantal Angueyra, http://orcid.org/0000-0002-2672-8249
Kamran Khodakhah, http://orcid.org/0000-0001-7905-5335

## Ethics
Animal experimentation: This study was performed in strict accordance with the recommendations in the Guide for the Care and Use of Laboratory Animals of the National Institutes of Health. All of the animals were handled according to approved institutional animal care and use committee (IACUC) protocols of the Albert Einstein College of Medicine. All surgery was performed under anesthesia, and every effort was made to minimize suffering. Protocol Number: 20160805

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
