## [Decision Letter]

Thank you for submitting your article "A role for cerebellum in the hereditary dystonia DYT1" for consideration by *eLife*. Your article has been favorably evaluated by Gary Westbrook (Senior Editor) and two reviewers, one of whom, Louis J Ptáček (Reviewer #1), is a member of our Board of Reviewing Editors. The reviewers have discussed the reviews with one another and the Reviewing Editor has drafted this decision to help you prepare a revised submission.

Summary:

The manuscript describes a mouse model of generalized dystonia using an shRNA knockdown approach. Although this disorder is the most common hereditary dystonia and the causative gene was identified almost 20 years ago, animal models have not been particularly reflective of the human disease. The human disorder has a remarkable developmental component. The disorder is only ~30% penetrant and if mutation carriers live to 30 years of age without developing symptoms, they will never develop generalized dystonia. It has been speculated that mouse models carrying germline mutations may have developmental compensation different from humans, which allows the mice to develop without showing signs of the disease.

The authors report that the cerebellum, not the basal ganglia, is the critical site for DYT1 knockdown, to trigger dystonia. Although this may seem logical and reasonable, especially in light of the authors' previous work in other models, it also challenges decades of focus on the basal ganglia in dystonia. The basal ganglia may still be involved and critical, but perhaps not the primary site where the gene causes physiological dysfunction. This discovery may refocus the field on understanding cerebellar circuitry and/or cerebellum-basal ganglia interactions. Finally, the authors provide some data regarding the potential mechanism, by establishing that PCs and DCN neurons have abnormal intrinsic excitability in slice preparations. Together, these findings are likely to have strong impact on the field of dystonia research, but also interest the field of motor control more broadly.

Essential revisions:

To strengthen the manuscript's novel findings, however, we recommend obtaining a few more N for key experiments and displaying additional data/analyses in the figures (see below).

1) Did the authors do physiology in the cerebellum in the animals with low% knockdown/no dystonia (i.e. animals to the lower left corner in Figure 1)? It would be interesting if the neurons recorded in less symptomatic animals had more normal activity patterns.

2) At the end of the Results section, the authors examine whether the mice show frank neurodegeneration, and find that there is indeed greater cell death in the shRNA knockdown animals compared to control in the DCN. Several questions arise: First, is it typical to get 5-10% of cells showing TUNEL staining in healthy control animals? Is the staining localized to the prior injection site, or is this a phenomenon that occurs around terminal procedures in the cerebellum? Also, for such an important finding (that there is indeed neurodegeneration in a DYT1 model mouse), more than N=3 mice per group should be used, and it would be nice to correlate the # of cells lost with the phenotype observed during life. It would be difficult to do the TUNEL staining, behavior, percent knockdown, and physiology in the same slice, but the individual correlations are interesting. Is the DCN like the SNc in parkinsonism, where the surviving cells develop abnormal discharge patterns?

---

## [Author Response]

*Essential revisions:*

*To strengthen the manuscript's novel findings, however, we recommend obtaining a few more N for key experiments and displaying additional data/analyses in the figures (see below).*

We agree that the number of animals reported in some experiments was indeed low. In some cases the experiments had already been done and we just needed to analyze the additional data to increase the N, and in other cases more experiments were done on animals that had been already injected with the shRNA.

*1) Did the authors do physiology in the cerebellum in the animals with low% knockdown/no dystonia (i.e. animals to the lower left corner in Figure 1)? It would be interesting if the neurons recorded in less symptomatic animals had more normal activity patterns.*

This is an interesting suggestion. In fact, our lab has extensive unpublished data performing extracellular awake recordings in several different mouse models of ataxia and dystonia of varying severity. What we find is that the irregularity of firing of cerebellar output indeed correlates linearly with the severity of the motor symptoms. This manuscript is next on our list of work to be submitted for publication.

However, to address the point raised by the reviewer in this manuscript we performed the experiments suggested in awake head-restrained torsinA KD mice which had a dystonia score between 1 and 2. These mice had motor dysfunction, but not dystonia. As expected, the firing properties of Purkinje cell and DCN neurons in these cells were, by and large, between those of the wild type and the dystonic mice. This data is now cited in the text and in Figure 2—figure supplement 1.

However, we caution that the interpretation of this data is difficult mainly because of the nature of the animal model. A mouse with a less severe motor symptom corresponds to one with less torsinA KD. This likely is the consequence of having fewer transfected cells, rather than less KD in each cell. In such a case the data from the mildly symptomatic mice might would be the average of firing properties of transfected and non-transfected cell, which is expected to fall in between the range of findings in for wildtype and the dystonic mice.

*2) At the end of the Results section, the authors examine whether the mice show frank neurodegeneration, and find that there is indeed greater cell death in the shRNA knockdown animals compared to control in the DCN. Several questions arise: First, is it typical to get 5-10% of cells showing TUNEL staining in healthy control animals? Is the staining localized to the prior injection site, or is this a phenomenon that occurs around terminal procedures in the cerebellum? Also, for such an important finding (that there is indeed neurodegeneration in a DYT1 model mouse), more than N=3 mice per group should be used, and it would be nice to correlate the # of cells lost with the phenotype observed during life.*

We agree with the reviewers, and therefore performed additional TUNEL experiments on 3 torsinA KD animals, 3 non-targeted injected animals, and 4 strain and age-matched wild-type animals. This data has now been added to Figure 4. The conclusion of the data remains the same, that there seems to be an increased susceptibility of the DCN neurons to torsinA KD.

We did find that there is a significant increase in cell death in non-targeted and torsinA injected mice when compared to wild-type animals, but that was restricted to the injection site and likely caused by injury. This data is presented in a revised Figure 4, and reviewers are invited to scrutinize panel C.

*It would be difficult to do the TUNEL staining, behavior, percent knockdown, and physiology in the same slice, but the individual correlations are interesting. Is the DCN like the SNc in parkinsonism, where the surviving cells develop abnormal discharge patterns?*

Yes, indeed. Our in vitro slice recordings (reported in Figure 3) were done in the presence of blocker of synaptic transmission, and we found that the intrinsic activity of both Purkinje cells and DCN neurons were altered. Thus, we found that DCN neurons in GFP positive areas of the slice fired abnormally with a decreased average firing rate and an increased CV ISI, whereas cells in non-GFP areas in the same slice fired normally, like the wild-type and non-targeted cells. Therefore, our interpretation of this data is that all infected cells with torsinA KD fire abnormally and after some time they begin to die. What triggers the cell death is unknown and we hope to use these animals in the future to study the mechanism underlying the irregular electrophysiological activity and the resulting cell death.